# Prevalence of hypertension and associated factors among nurses working in private and public tertiary hospitals in Dar Es Salaam, Tanzania

Jestina M. Rutagengwa[1]*, Theresia A. Ottaru[2], Ezra J. Mrema[1], Zuhura I. Kimera[1], Hussein H. Mwanga[1]

**1** Department of Environmental and Occupational Health, School of Public Health and Social Sciences, Muhimbili University of Health and Allied Sciences, United Nations Road, Upanga West, Dar es Salaam, Tanzania, **2** Department of Epidemiology and Biostatistics, School of Public Health and Social Sciences, Muhimbili University of Health and Allied Sciences, United Nations Road, Dar es Salaam, Tanzania

* jestinarutagengwa@gmail.com

## Abstract

Nurses face numerous occupational stressors associated with an increased risk of hypertension. This study aimed to determine the prevalence of hypertension and its associated factors among nurses in private and public tertiary hospitals in Dar es Salaam, Tanzania. A cross-sectional study was conducted between July and November 2023 among nurses from one private and one public tertiary hospital. All eligible nurses in the private hospital were recruited, while stratified random sampling was used to recruit nurses at the public hospital. Data were collected using questionnaires and physical measurements (blood pressure, weight, and height). Hypertension was defined as blood pressure ≥130/80 mmHg, self-reported history of hypertension, or current use of antihypertensive medication. Descriptive statistics summarized participants' characteristics, and logistic regression analysis identified factors independently associated with hypertension. Among 360 nurses [median age 29 years (IQR 27–31), 60% < 35 years, 57.5% female], 28.9% were hypertensive. After adjusting for age, sex, and high cholesterol, several factors were independently associated with hypertension. Nurses aged ≥55 years had significantly higher odds of hypertension (AOR 9.20; 95% CI:5.90–14.56). Higher educational attainment (AOR 5.20; 95% CI:2.49–11.01), being unmarried (AOR 1.91; 95% CI:1.09–3.36), physical inactivity (AOR 3.23; 95% CI:2.39–15.60), overweight (AOR 2.73; 95% CI:1.29–3.75), obesity (AOR 3.30; 95% CI:3.13–13.05), poor dietary practices (AOR 2.73; 95% CI:1.29–13.75), and short sleep duration (≤5 hours) (AOR 4.20; 95% CI:1.06–7.73) were significantly associated with hypertension. Work-related stress (AOR 2.86; 95% CI:1.06–7.73) and attending to >15 patients per day (AOR 3.67; 95% CI:1.32–5.40) were also associated with increased risk. The study revealed a significant prevalence of hypertension among nurses, associated with behavioral

**Data availability statement:** An anonymized dataset containing individual-level data (demographics, blood pressure, behavioral and occupational variables) is deposited in a public repository (Figshare) DOI; 10.6084/m9.figshare.30648137.

**Funding:** The authors received no specific funding for this work.

**Competing interests:** The authors have declared that no competing interests exist.

and work-related factors. Interventions promoting healthy diets, physical activity, and stress management at the workplace are recommended to safeguard nurses' health and improve healthcare delivery.

## Introduction

Hypertension, or high blood pressure, is a chronic medical condition characterized by prolonged elevated arterial pressure [1]. It is influenced by several factors, including poor dietary habits, physical inactivity, advancing age, and biological sex. Globally, hypertension remains a leading contributor of non-communicable diseases (NCDs) and is projected to affect approximately 1.6 billion adults by 2025 [2]. In Sub-Saharan Africa, out of approximately 650 million people, between 10 and 20 million have hypertension. Its prevalence is projected to increase to 21.8 million by 2030 [2]. In Tanzania, the prevalence of hypertension in the general population ranges between 28% and 40% [3].

The high burden of hypertension poses serious public health challenges, especially in low- and middle-income countries, where it contributes significantly to morbidity, mortality, and increased healthcare costs [4]. Risk factors for hypertension are classified as modifiable, such as unhealthy diet, physical inactivity, tobacco use, and excessive alcohol intake, and non-modifiable, including age, sex, and genetic factors [5]. Evidence also suggests higher odds of hypertension among males and individuals with limited knowledge about the condition [6].

Importantly, these factors may vary by occupational group. Among healthcare workers, particularly doctors and nurses, occupational stressors such as long working hours, shift work and job-related stress may further elevate the risk of developing hypertension [7]. Despite nurses playing a critical role in the prevention and management of hypertension, their health, particularly regarding hypertension, has received limited attention; a study by Guwatudde *et al.* [8] indicated a prevalence of 25% among nurses in Tanzania, Uganda, Nigeria, and South Africa.

Contrary to expectations, hypertension among nurses is not necessarily lower than in the general population. Recent studies have shown that poor lifestyle choices, high body weight, and alcohol use are prevalent in this group, compounding their occupational risks [8,9]. These factors make nurses a unique and important population for studying hypertension-related risk.

This study, therefore, aimed to assess the prevalence of hypertension and its associated factors among nurses in tertiary hospitals in Dar es Salaam, Tanzania. The findings highlight the burden of hypertension among this critical workforce and inform strategies to reduce modifiable risk factors.

## Materials and methods

### Ethics statement

Ethical approval was obtained from the Institutional Review Board of Muhimbili University of Health and Allied Sciences (Reference No: DA.282/298/01.C/1817,

July 2023). Participants were fully informed about the study's purpose and provided written informed consent. Participation was voluntary, and respondents could withdraw at any time. Data were anonymized and kept confidential to ensure privacy and ethical integrity.

## Study design

This was a hospital-based cross-sectional analytical study conducted among nurses employed in both private (n = 1) and public (n = 1) tertiary hospitals in Dar es Salaam, Tanzania to assess the prevalence of hypertension and its associated factors.

## Study area

The study was conducted in two tertiary hospitals located in Dar es Salaam, Tanzania. The public hospital is a national tertiary referral, research, and university teaching hospital with a capacity of 1,500 beds. It serves over 2,000 outpatients daily, employs approximately 2,800 staff, and offers a wide range of services including education, screening, diagnosis, and treatment. The hospital receives referrals from across the country, often managing complex cases and serving patients from lower socio-economic backgrounds. The private hospital is a multi-specialist tertiary facility with a capacity of 170 beds and about 500 healthcare workers across more than 10 departments. It provides general medical services, specialist clinics, and diagnostic care, and is one of the tertiary referral centers receiving patients from various regions within Tanzania as well as from neighboring countries.

## Study population, sample size and sampling strategy

The study population consisted of permanently employed nurses from both hospitals, including 190 nurses from the private hospital and 1,105 nurses from the public hospital. The sample size was estimated using Cochran's formula, with the prevalence of 40% for hypertension among healthcare workers [9], at 95% confidence level and 5% margin of error. The two hospitals were purposively selected to represent private and public tertiary-level healthcare institutions.

In the public hospital, seven departments with more than ten nurses were purposively selected: the Medical Ward, VIP Ward, Pediatric Ward, Surgical Ward, Neonatal Ward, Obstetrics (Maternity) and Gynecology Ward. Stratified random sampling was then used to recruit participants from each department, with the number of nurses selected proportionate to the size of the department.

In the private hospital, due to the relatively small number of nurses, all who met the inclusion criteria (i.e., all nurses working at the hospital at the time and are willing to participate) were invited to participate. Nurses who were pregnant or unavailable during the data collection period were excluded from the study.

## Data collection

Participant recruitment for this study commenced on 27/07/2023 and concluded on 30/09/2023, during which each participant provided written informed consent prior to participation. Each participant responded to a questionnaire that collected information on sociodemographic characteristics, behavioral factors, and work-related factors. Responses were regularly reviewed for completeness and accuracy. Blood pressure was measured three times using a calibrated digital sphygmomanometer, with five-minute intervals between measurements following AHA recommendations [1]. The average of the last two readings was used. A participant was considered hypertensive if they met any of the following: (i) Blood pressure ≥ 130/80 mmHg, (ii) Self-reported history of hypertension, or (iii) Current use of antihypertensive medication.

The same questionnaire was used to gather information on clinical factors, including a history of comorbidities such as diabetes mellitus, and hypercholesterolemia. Participants were asked whether they had ever been diagnosed with these conditions by a medical professional or were currently taking prescribed medications for them, and they were considered to have a comorbid if they answered "yes".

Behavioral factors assessed included dietary habits, physical activity, alcohol consumption, nutritional status, and cigarette smoking. Dietary habits were categorized into good or poor. Participants were classified as having good dietary habits if they consumed at least one serving of fruits and vegetables daily on more than three days per week and consumed less than 2 grams of salt per day.

The Body Mass Index (BMI) was calculated from height and weight measurements according to WHO guidelines [10]. Height was measured with the participants standing on a portable stadiometer with bare feet and light clothing. Weight was measured to the nearest 100g by using an electronic weighing scale. Participants were categorized as underweight ($<18$ kg/m$^2$), normal (18.5 kg/m$^2$), overweight (25.0–29.9 kg/m²), or obese ($\geq30$ kg/m²).

Physical activity was assessed using a modified WHO STEPwise approach (STEP) tool [11]. Participants were classified as physically active if they engaged in moderate to vigorous physical activity for at least 10 minutes per session, on 3–5 days per week. Moderate physical activity was defined as exercise which causes a noticeable but manageable increase in breathing or heart rate, such as brisk walking, household chores, or gardening. Vigorous physical activity was defined as exercise which results in a substantial increase in heart rate and breathing, including running, aerobics, or heavy lifting [12].

Participants were categorized as excessive drinkers based on the number of drinks per day. Excessive consumption was defined as having > 2 drinks per day for men and > 1 drink per day for women. Cigarette smoking was categorized as either current smokers or nonsmokers.

Data on work-related factors such as working hours, sleep duration, shift schedules, and perceptions of the work environment were collected. Work-related stress was measured using the Workplace Stress Scale, developed by the American Institute of Stress, which categorizes stress levels into calm, moderate, severe, and potentially dangerous categories. This 40-point scale has previously been used to assess occupational stress among healthcare workers [13].

Depression levels were evaluated using the PHQ-9 scale, which assesses the presence and severity of depressive symptoms. The PHQ-9 has a maximum score of 27, with scores categorized as follows: 0–4 indicating no depression, 5–9 mild depression, 10–14 moderate depression, 15–19 moderately severe depression, and 20–27 severe depression [14]. Sleep duration was collected as a continuous variable and then categorized as ≤5 hours (very poor), 6–7 hours (poor), and >7 hours (good) per night, based on prior studies linking short sleep to adverse health outcomes such as hypertension and metabolic disorders [15].

## Data analysis

Data were analyzed using the Statistical Package for Social Sciences (SPSS) version 25. Descriptive statistics were performed to summarize the characteristics of the study population: means and standard deviations for continuous variables, and frequencies and percentages for categorical variables, stratified by health facility type and hypertension status. Bivariate logistic regression analysis was initially performed to examine the association between hypertension and individual sociodemographic, clinical, behavioral, and work-related factors. Multivariable logistic regression models, adjusted for age, sex and high cholesterol, were then used to assess these associations. Results were reported as adjusted odds ratios (AOR) with 95% confidence intervals (CI).

## Results

### Characteristics of the study participants

A total of 360 nurses participated in the study, with a median age of 29 years (IQR 27 − 31), drawn from both private and public tertiary hospitals (Table 1). The majority of the study participants (60%) were younger than 35 years. The overall prevalence of hypertension among nurses was 28.9%. Among the 104 (28.9%) nurses identified as hypertensive, 38 (36.5%) were newly identified during screening using the sphygmomanometer while 66 (63.5%) reported a prior diagnosis

**Table 1. Sociodemographic and clinical characteristics of the study participants (n = 360).**

| Variables | Health facilities | | | |
|---|---|---|---|---|
| | N = 360 | Public hospital n = 204 (%) | Private hospital n = 156 (%) | p-value (Chi-squared test) |
| **Age** (years): Median (IQR) | 29 (27 – 31 ) | 28 (27 –30) | 30 (28 –31) | 0.450* |
| <35 | 216 (60.0) | 114 (55.9) | 102 (65.4) | 0.006 |
| 35-44 | 78 (21.6) | 48 (23.5) | 30 (19.2) | |
| 45-54 | 33 (9.2) | 27 (13.2) | 6 (3.9) | |
| ≥55 | 33 (9.2) | 15 (7.4) | 18 (11.5) | |
| **Sex** | | | | 0.949 |
| Male | 153 (42.5) | 87 (42.6) | 66 (42.3) | |
| Female | 207 (57.5) | 117 (57.4) | 90 (57.7) | |
| **Marital Status** | | | | 0.037 |
| Married | 168 (46.7) | 105 (51.5) | 63 (40.4) | |
| Not married | 192 (53.3) | 99 (48.5) | 93 (59.6) | |
| **Level of Education** | | | | 0.070 |
| Certificate to diploma | 195 (54.2) | 119 (58.3) | 76 (48.7) | |
| Degree and beyond | 165 (45.8) | 85 (41.7) | 80 (51.3) | |
| **Family History of Hypertension** | | | | 0.068 |
| Yes | 144 (40.0) | 90 (44.1) | 54 (34.6) | |
| No | 216 (60.0) | 114 (55.9) | 102 (65.4) | |
| **Comorbidities** | | | | 0.589 |
| None | 285 (79.2) | 165 (80.9) | 120 (76.9) | |
| Diabetes | 27 (7.5) | 15 (7.4) | 12 (7.7) | |
| High cholesterol | 48 (13.3) | 24 (11.7) | 24 (15.4) | |
| **Hypertension** | | | | 0.036 |
| Yes | 104 (28.9) | 50 (24.5) | 54 (34.6) | |
| No | 256 (71.1) | 154 (75.5) | 102 (65.4) | |

*: Wilcoxon sum-rank test; IQR: Interquartile Range

of hypertension. Among those with a prior diagnosis of hypertension, 45 (68.2%) were currently using antihypertensive medications.

A higher prevalence of hypertension was observed among those working in public hospitals compared to private hospital (31.4% vs 25.6%, p = 0.04). Approximately 40% of respondents from both institutions reported a family history of hypertension. Additionally, elevated cholesterol levels were more frequently reported among nurses from the private hospital (15.7%) compared to those from the public hospital (11.7%).

## Sociodemographic and clinical factors associated with hypertension

In the multivariable regression analysis, several factors were significantly associated with hypertension among nurses in Dar es Salaam (Table 2). Nurses aged ≥55 years had higher odds of hypertension (AOR = 9.20; 95% CI: 5.90–14.56) compared to those under 35 years. Similarly, nurses with a degree or higher education level had increased odds of hypertension (AOR = 5.20; 95% CI: 2.49–11.01) relative to those with a certificate or diploma. Unmarried nurses were also more likely to be hypertensive (AOR = 1.91; 95% CI: 1.09–3.36) compared to their married counterparts. Furthermore, hypertensive nurses were more likely to be overweight (AOR = 1.13; 95% CI: 1.01 – 2.45) or obese (AOR = 2.54;

**Table 2. Sociodemographic and clinical factors associated with hypertension among nurses (n = 360).**

| Variable | | Hypertension status | | Multivariable logistic analysis |
|---|---|---|---|---|
| | N = 360 (%) | Hypertensive n = 104, 28.9% n (%) | Non-Hypertensive n = 256, 71.1% n (%) | AOR (95% CI) |
| **Age** (years)¤ | | | | |
| ≥ 55 | 33 (9.2) | 12 (36.4) | 21 (63.6) | 9.20 (5.90 - 14.56)* |
| 45 – 54 | 33 (9.2) | 15 (45.5) | 18 (54.5) | 0.28 (0.21 - 0.69) |
| 35 – 44 | 78 (21.6) | 30 (38.5) | 48 (61.5) | 0.24 (0.11 - 0.47) |
| < 35 | 216 (60.0) | 47 (21.8) | 169 (78.2) | *Ref* |
| **Sex**¶ | | | | |
| Female | 207 (57.5) | 64 (30.9) | 143 (69.1) | 0.88 (0.50 - 1.55) |
| Male | 153 (42.5) | 40 (26.1) | 113 (73.9) | *Ref* |
| **Education** | | | | |
| Certificate to diploma | 195 (54.2) | 70 (35.9) | 125 (64.1) | 5.20 (2.49 - 11.01)* |
| Degree and beyond | 165 (45.8) | 34 (20.6) | 131 (79.4) | *Ref* |
| **Marital status** | | | | |
| Not married | 192 (53.3) | 47 (24.5) | 145 (75.5) | 1.91 (1.09 - 3.36)* |
| Married | 168 (46.7) | 57 (33.9) | 111 (66.1) | *Ref* |
| **History of Comorbidities** | | | | |
| Diabetes | 27 (7.5) | 9 (33.3) | 18 (66.7) | 0.89 (0.70- 1.12) |
| High cholesterol | 48 (13.3) | 28 (58.3) | 20 (41.7) | 0.73 (0.12 - 7.75) |
| None | 285 (79.2) | 67 (23.5) | 218 (76.5) | *Ref* |
| **Body Mass Index (BMI)** | | | | |
| Obese (≥ 30 kg/m²) | 93 (25.8) | 42 (45.2) | 51 (54.8) | 2.54 (1.15 - 5.62)* |
| Overweight (25.0–29.9 kg/m²) | 145 (40.3) | 34 (23.5) | 111 (76.5) | 1.13 (1.01 – 2.45)* |
| Underweight (<18.5 kg/m²) | 6 (1.7) | 1 (16.7) | 5 (83.3) | 0.91 (0.06 - 7.38) |
| Normal (18.5 – 24.9 kg/m²) | 116 (32.2) | 27 (23.3) | 89 (76.7) | *Ref* |
| **Family History of Hypertension** | | | | |
| Yes | 144 (40.0) | 45 (31.2) | 99 (68.8) | 2.08 (1.19 - 3.64)* |
| No | 216 (60.0) | 59 (27.3) | 157 (72.7) | *Ref* |

*AOR: Adjusted odds ratio; CI: Confidence interval; Ref: Reference category; Each AOR represents a separate model adjusted for age, sex, and high cholesterol, unless otherwise specified; €: Adjusted for sex and high cholesterol;*

*¶: Adjusted for age and high cholesterol; *: p-value < 0.05*

95% CI = 1.15-5.62). A family history of hypertension was associated with two times higher odds of being hypertensive (AOR = 2.08; 95% CI: 1.19–3.64).

### Behavioral factors associated with hypertension

Several behavioral factors were significantly associated with hypertension among nurses in the multivariable analysis (Table 3). Nurses who were less physically active had a significantly higher odds of hypertension compared to their more active counterparts (AOR = 3.23; 95% CI: 2.39–15.6). Furthermore, hypertensive nurses were more likely to have poor dietary habits (AOR = 2.73; 95% CI: 1.29-3.75). Nurses experiencing less than 5 hours of sleep showed significantly higher odds of being hypertensive (AOR = 4.20; 95% CI: 1.06-7.73) compared to those who had more than 7 hours of sleep. In contrast, smoking and alcohol use did not show a significant association with hypertension in this analysis.

**Table 3. Behavioral factors associated with hypertension among nurses (n = 360).**

| Variables | | Hypertension status | | Multivariable logistic analysis |
|---|---|---|---|---|
| | N = 360 (%) | Hypertensive n=104, 28.9% n (%) | Non-Hypertensive n=256, 71.1% n (%) | AOR (95% CI) |
| **Physical Activities** | | | | |
| Less Active | 152 (42.2) | 48 (31.6) | 104 (68.4) | 3.23 (2.39 - 15.6)* |
| Active | 208 (57.8) | 56 (26.9) | 152 (73.1) | *Ref* |
| **Smoking Habit** | | | | |
| Yes | 40 (11.1) | 9 (22.5) | 31 (77.5) | 0.73 (0.29 - 3.75) |
| No | 320 (88.9) | 95 (29.7) | 225 (70.3) | *Ref* |
| **Excessive alcohol use** | | | | |
| Yes | 145 (40.3) | 46 (31.7) | 99 (68.3) | 0.63 (0.12 - 7.75) |
| No | 215 (59.7) | 58 (27.0) | 157 (73.0) | *Ref* |
| **Dietary Practices** | | | | |
| Poor | 310 (86.1) | 62 (20.0) | 248 (80.0) | 2.73 (1.29 - 3.75)* |
| Good | 50 (13.9) | 42 (84.0) | 8 (16.0) | *Ref* |
| **Sleep duration** (hours) | | | | |
| ≤ 5 | 94 (26.1) | 60 (63.8) | 34 (36.2) | 4.20 (1.06 - 7.73)* |
| 6 – 7 | 115 (31.9) | 25 (21.7) | 90 (78.3) | 2.30 (0.32 - 3.40) |
| > 7 | 151 (42.0) | 19 (12.6) | 132 (87.4) | *Ref* |

*AOR: Adjusted odds ratio; CI: Confidence interval; Ref: Reference category; BMI: Body Mass Index; kg: kilogram; m: meter; Each AOR represents a separate model adjusted for age, sex, and high cholesterol; *: p-value < 0.05.*

### Work-related factors associated with hypertension

After adjusting for age, sex, and high cholesterol, several work-related factors were found to be associated with hypertension among this group of nurses (Table 4). Nurses who reported experiencing potentially dangerous or severe work-related stress had significantly higher odds of hypertension compared to those who described their work-related stress as relatively calm (AOR = 2.86; 95% CI: 1.067.73). Patient load was also a significant factor; nurses attending to more than 15 patients per day had two-fold higher odds of being hypertensive (AOR = 2.20; 95% CI:0.88- 5.45), compared to those attending to only 1–5 patients daily. Other work-related factors, such as workplace setting (public *vs.* private hospital), number of working days per week, depression status, and presence of wellness programs were not significantly associated with hypertension in this study.

## Discussion

This study assessed the prevalence and associated factors of hypertension among nurses working in both private and public tertiary hospitals in Dar es Salaam, Tanzania. The findings revealed that hypertension is relatively prevalent within this professional group, with a rate of 28.9%, which is slightly higher than the national prevalence of 26% in the general Tanzanian population [13]. Several factors were significantly associated with hypertension: advanced age, lower education level, unmarried status, a family history of hypertension, physical inactivity, overweight and obesity, poor dietary habits, sleep deprivation, work-related stress, and high patient workload.

The observed hypertension prevalence among nurses in this study aligns with findings from other sub-Saharan African countries. For instance, a study conducted among nurses in Nigeria reported an age-adjusted hypertension prevalence of 25.8%, while school teachers in Tanzania exhibited a prevalence of 23.1%. These figures are comparable to the 28.9%

**Table 4. Work-related factors associated with hypertension among nurses (n = 360).**

| Variables | | Hypertension status | | Multivariable logistic analysis |
|---|---|---|---|---|
| | N = 360 (%) | Hypertensive n = 104, 28.9% n (%) | Non-Hypertensive n = 256, 71.1% n (%) | AOR (95% CI) |
| **Work-related stress** | | | | |
| Potentially dangerous or Severe Stress | 35 (9.7) | 9 (25.7) | 26 (74.3) | 2.86 (1.06 - 7.73)* |
| Moderate stress | 71 (19.7) | 33 (46.5) | 38 (53.5) | 1.36 (0.48 - 3.86) |
| Fairly low | 86 (23.9) | 20 (23.3) | 66 (76.7) | 0.67 (0.32 - 1.40) |
| Relative calm | 168 (46.7) | 42 (25.0) | 126 (75.0) | *Ref* |
| **Patients attended per day** | | | | |
| > 15 | 60 (16.7) | 18 (30.0) | 42 (70.0) | 2.20 (0.88- 5.45)* |
| 6–15 | 160 (44.4) | 43 (26.9) | 117 (73.1) | 0.78 (0.42 - 1.41) |
| 1–5 | 140 (38.9) | 43 (30.7) | 97 (69.3) | *Ref* |
| **Workplace** | | | | |
| Private hospital | 156 (43.3) | 54 (34.6) | 102 (65.4) | 0.72 (0.12 - 3.40) |
| Public hospital | 204 (56.7) | 50 (24.5) | 154 (75.5) | *Ref* |
| **Working days** | | | | |
| >3 | 237 (65.8) | 69 (29.1) | 168 (70.9) | 1.58 (0.85 - 2.89) |
| ≤3 | 123 (34.2) | 35 (28.5) | 88 (71.5) | *Ref* |
| **Depression status** | | | | |
| Moderate to Severe | 44 (12.2) | 17 (38.6) | 27 (61.4) | 2.35 (1.21 - 4.60)* |
| Mild | 115 (31.9) | 36 (31.3) | 79 (68.7) | 1.00 (0.01 - 1.20) |
| None | 201 (55.8) | 51 (25.4) | 150 (74.6) | *Ref* |
| **Presence of a wellness program** | | | | |
| No | 217 (60.3) | 47 (21.7) | 170 (78.3) | 0.56 (0.01 - 5.20) |
| Yes | 143 (39.7) | 57 (39.9) | 86 (60.1) | *Ref* |

*AOR: Adjusted odds ratio; CI: Confidence interval; Ref: Reference category; Each AOR represents a separate model adjusted for age, sex, and high cholesterol; *: p-value < 0.05; **: p-value < 0.01; ***: p-value < 0.001.*

prevalence found in our study, underscoring the occupational health risks faced by healthcare professionals in the region [8]. Notably, the study revealed a slightly higher prevalence of hypertension among nurses working in public hospitals compared to those in private settings. This disparity may be attributed to differences in workload, institutional infrastructure, and shift schedules between public and private healthcare facilities. Similar patterns have been observed in other professions; for example, a study by Baklouti *et al.* [16] found higher hypertension rates among public school teachers compared to their counterparts in private schools. These findings suggest that organizational factors and work environments play a significant role in the health outcomes of employees.

Age emerged as a strong predictor of hypertension in this population. Nurses aged ≥55 years had significantly higher odds of being hypertensive compared to younger nurses. This finding is consistent with global evidence demonstrating a strong correlation between aging and increased blood pressure due to arterial stiffening, endothelial dysfunction, and accumulated exposure to behavioral risk factors over time [17]. Additionally, with career progression and longer service duration, nurses may experience more intense work-related stress and physical demands, contributing to elevated blood pressure [18].

Educational attainment was also significantly associated with hypertension. Nurses with only certificate or diploma-level education had higher odds of hypertension compared to those with a degree or higher. Similar findings were reported in

studies conducted in Ghana and Italy, which linked lower educational levels to limited health literacy, unhealthy lifestyle behaviors, and inadequate stress management factors commonly associated with lower socioeconomic status [19,20]. Addo *et al.* (2012) further emphasized that individuals with higher education were more likely to engage in health-promoting behaviors and adhere to treatment regimens [21].

Marital status was another significant factor, with unmarried nurses exhibiting higher rates of hypertension. Previous research has demonstrated that being married or in a stable relationship provides emotional and social support that can buffer against chronic stress and its physiological consequences [22–24]. These protective effects may contribute to healthier lifestyle behaviors and better cardiovascular outcomes in married individuals. Interestingly, despite the majority of the study participants being female, gender was not significantly associated with hypertension. This contrasts with studies that have linked hormonal changes, particularly post-menopause, to increased hypertension risk in women [18]. However, occupational demands and psychosocial stress may override gender-related physiological differences in this context. A study in Ethiopia similarly found no significant gender difference among healthcare professionals, with age, physical inactivity, and poor dietary adherence being the dominant predictors [25].

As expected, a family history of hypertension significantly increased the risk, with nurses reporting such a history being over twice as likely to be hypertensive. This aligns with global literature recognizing family history as a strong, non-modifiable risk factor, attributable to both genetic predisposition and shared lifestyle environments [26,27]. Family history is often correlated with obesity, metabolic syndrome, and early-onset hypertension, especially in populations with limited preventive care access [28].

Body Mass Index (BMI) was one of the strongest behavioral predictors of hypertension. Both overweight and obesity were significantly associated with increased odds of hypertension. These findings are consistent with those of Okelo *et al.,* who also reported higher mean systolic blood pressure among overweight and obese participants. In South Africa, a similar study found that 73% of healthcare workers were overweight or obese, with cultural norms reinforcing the perception of larger body size as a sign of health and wealth [29]. Additionally, increasing availability of energy-dense foods and declining physical activity in urban settings contribute to the rising prevalence of obesity [30].

Dietary behavior further explained hypertension risk. Poor dietary practices were common, particularly among nurses in public hospitals (61.8%), and were linked to higher rates of hypertension. Long shifts, fatigue, and lack of structured wellness programs often limit nurses' ability to maintain healthy eating habits [31]. A systematic review suggests that hospital-based interventions focused on dietary and physical activity support can significantly improve health outcomes among healthcare workers [32].

Physical inactivity was another modifiable risk factor identified in the study. Less active nurses had over threefold increased odds of being hypertensive compared to their more active counterparts. This finding aligns with studies from India and South Africa, where shift work and extended hours were shown to limit physical activity, increasing the risk of obesity and related conditions [17,24]. Nurses are particularly vulnerable due to the nature of their work, which often involves inconsistent schedules, high physical demands, and insufficient time for exercise.

Although smoking and alcohol use were not significantly associated with hypertension in this study, their potential role cannot be overlooked. Other studies have demonstrated that smoking, in particular, is a key contributor to elevated blood pressure among healthcare workers, particularly male nurses [4,33]. The low smoking prevalence in this sample may reflect effective anti-smoking policies within Tanzanian healthcare institutions or underreporting due to social desirability bias.

Sleep duration was significantly associated with hypertension, with individuals reporting shorter sleep hours exhibiting a higher risk of elevated blood pressure. This finding suggests that sleep deprivation, often compounded by long working hours, may increase stress levels, which in turn contributes to the development of hypertension. Physiologically, restricted sleep can lead to heightened sympathetic nervous system activity, elevated cortisol levels, endothelial dysfunction, and impaired glucose metabolism, all of which are established mechanisms for hypertension [15]. These results are consistent

with previous studies reporting a strong link between short sleep duration and increased hypertension risk, for instance, a study done by Phiri et al. (2014) showed that short duration of sleep was associated with a greater risk of developing or dying of cardiovascular diseases such as hypertension [34].

Occupational factors such as workload, stress and working more than 3 days per week were also strongly associated with hypertension. Nurses attending to more than 15 patients per day had two-fold increased odds of hypertension, highlighting the critical imbalance in nurse-to-patient ratios, especially in public hospitals. This heavy workload is a known contributor to both physical fatigue and psychological stress, potentially exacerbating hypertension risk [24]. The recommended nurse-to-population ratio for achieving Universal Health Coverage (UHC) is at least 83 nurses per 10,000 people, yet this standard is often unmet in resource-limited settings. Most sub-Saharan African countries have fewer than 20 nurses per 10,000 patients or 2 nurses per 1,000 patients, reflecting a significant shortage [35].

Nurses who worked more than three days per week were 1.58 times more likely to have hypertension compared to those working fewer days. Increased workdays may contribute to prolonged occupational stress, reduced recovery time, and physiological strain, all of which can heighten the risk of elevated blood pressure. Similar findings have been reported in studies conducted among healthcare workers, where longer work schedules and inadequate rest periods were associated with higher rates of hypertension and cardiovascular risk [36].Evidence from a cohort study among shift workers also demonstrated that prolonged weekly work hours and insufficient rest between shifts contribute to sympathetic over activity, sleep disturbances, and metabolic dysregulation, ultimately increasing hypertension risk [37].

Finally, nurses reporting high work-related stress were twice as likely to have hypertension. This supports existing literature that links chronic stress, especially in high-pressure units like emergency or ICU to increased cardiovascular risk [23]. The combined effects of shift work, long hours, high patient volumes, and lack of control over work conditions can contribute to both physical and emotional strain, further amplifying stress levels.

## Limitations

Some study limitations should be considered when interpreting the findings. First, because this was a cross-sectional design, it cannot establish causal relationships between the identified factors and hypertension. Second, the study was conducted in only two tertiary hospitals located in urban Dar es Salaam, which may limit the generalizability of the findings to nurses working in rural or lower-level health facilities. Third, the use of self-reported information may have introduced recall and social desirability bias; however, such biases are likely minimal since standardized and validated tools were used. Additionally, some associations showed wide confidence intervals, likely due to smaller subgroup sizes and uneven exposure distributions. Nevertheless, the regression models were assessed for fit and multicollinearity, and no evidence of model instability was observed. Thus, the results should be interpreted with caution.

## Conclusion

This study revealed a high prevalence of hypertension among nurses working in tertiary hospitals in Dar es Salaam, indicating a substantial occupational health concern. The findings identified associations between hypertension and several sociodemographic, behavioral, and work-related factors, such as advanced age, unmarried status, physical inactivity, overweight and obesity, poor dietary habits, inadequate sleep, work-related stress, and heavy workloads. The findings underscore the importance of workplace-based interventions that promote healthy lifestyle behaviors, stress management, and supportive work environments. Future studies involving larger and more diverse samples are recommended to further explore these associations and guide broader occupational health policies.

## Acknowledgments

We sincerely thank all the nurses from Muhimbili National Hospital and Aga Khan Hospital who generously consented to participate in this study

## Author contributions

**Conceptualization:** Jestina Modest Rutagengwa.

**Data curation:** Jestina Modest Rutagengwa, Theresia A. Ottaru.

**Formal analysis:** Jestina Modest Rutagengwa.

**Investigation:** Jestina Modest Rutagengwa.

**Methodology:** Jestina Modest Rutagengwa.

**Resources:** Jestina Modest Rutagengwa.

**Supervision:** Theresia A. Ottaru, Ezra J. Mrema, Zuhura I. Kimera, Hussein H. Mwanga.

**Validation:** Ezra J. Mrema, Zuhura I. Kimera, Hussein H. Mwanga.

**Visualization:** Ezra J. Mrema, Zuhura I. Kimera, Hussein H. Mwanga.

**Writing – original draft:** Jestina Modest Rutagengwa.

**Writing – review & editing:** Jestina Modest Rutagengwa, Hussein H. Mwanga.

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
