## [Decision Letter · Decision Letter 0]

20 Oct 2025

PGPH-D-25-02168

Prevalence of Hypertension and Associated Factors among Nurses Working in Private and Public Tertiary Hospitals in Dar Es Salaam, Tanzania

Dear Dr. Rutagengwa,

Thank you for submitting your manuscript to PLOS Global Public Health. After careful consideration, we feel that it has merit but does not fully meet PLOS Global Public Health’s publication criteria as it currently stands. Therefore, we invite you to submit a revised version of the manuscript that addresses the points raised during the review process.

This is a well written manuscript which is very relevant in the context of the increased prevalence of Malaria where a rapid diagnosis is imperative. To add onto the good efforts of the authors the following points are suggested

Please edit the repetitions in sentences and interpretations so that the number of words in abstract will be within the stipulated limits and  the manuscript gives more clarity.The methods training for the  health care workers and the policy implications may be describedThe limitation of the study also need to be elaborated

We look forward to receiving your revised manuscript.

Kind regards,

Suma Krishnasastry, MBBS, MD,DNB, FRCP (Edin)

Academic Editor

Journal Requirements:

1. Please ensure that your Ethics Statement is available in its entirety at the beginning of your Methods section, under a subheading 'Ethics Statement'.

2. We note that your Data Availability Statement is currently as follows: “All relevant data are fully contained within the manuscript. No additional datasets were generated or analyzed during the current study."

Additional Editor Comments (if provided):

Reviewers' comments:

Reviewer's Responses to Questions

**Comments to the Author**

1. Does this manuscript meet PLOS Global Public Health’s publication criteria?

Reviewer #1: Yes

Reviewer #2: Yes

Reviewer #3: Partly

Reviewer #4: Yes

2. Has the statistical analysis been performed appropriately and rigorously?

Reviewer #1: Yes

Reviewer #2: Yes

Reviewer #3: No

Reviewer #4: Yes

3. Have the authors made all data underlying the findings in their manuscript fully available (please refer to the Data Availability Statement at the start of the manuscript PDF file)?

Reviewer #1: Yes

Reviewer #2: Yes

Reviewer #3: No

Reviewer #4: No

4. Is the manuscript presented in an intelligible fashion and written in standard English?

Reviewer #1: Yes

Reviewer #2: Yes

Reviewer #3: Yes

Reviewer #4: Yes

Reviewer #1: This is an interesting article about factors associated with Hypertension in this special group. The title and abstract in a clear sentences but abstract must not be more than 300, precisely describe the main aspect of this study.

The introduction is well composed, updated and informative, mentioning recent publications about the problem of investigation.

Methods are clearly described

Results are important and relevant,

Discussion is clear, well-structured and designed, correlates with the results.

Conclusions correlate to the results

The figures and tables are clear and legible

Reviewer #2: I find this to be a well-structured and relevant cross-sectional study that addresses an important occupational health issue in a low-resource setting. The manuscript is clearly written, methods are appropriately described, and the sampling strategy is justified given the context. The prevalence of hypertension (28.9%) among nurses is concerning and highlights a significant public health problem. The use of multivariable regression to identify associated factors is a strength, and several findings—such as the strong associations with age, BMI, physical inactivity, and work-related stress—are consistent with existing literature. However, the study is limited by its focus on only two urban tertiary hospitals, which affects generalizability, and the reliance on self-reported data for behavioral factors introduces potential bias. Additionally, some confidence intervals in the results are unusually wide (e.g., for physical inactivity and poor diet), which may reflect small subgroup sizes or model instability and should be interpreted with caution. Despite these limitations, the study provides valuable insights and practical recommendations for workplace interventions aimed at reducing hypertension risk among nurses in Tanzania.

Reviewer #3: This manuscript explores the prevalence of hypertension among nurses in public and private tertiary hospitals in Dar es Salaam, Tanzania, and examines sociodemographic, behavioral, and occupational risk factors. The topic is important, timely, and relevant for occupational and public health in sub-Saharan Africa. The use of standardized blood pressure measurement methods is a strength, and the inclusion of both private and public hospital settings adds useful context. However, several aspects of the study require clarification or revision before the manuscript can be considered for publication.

Methodology and Analysis: The cross-sectional design is appropriate for estimating prevalence but does not allow for causal inference. This limitation should be stated more clearly in the discussion. Logistic regression is correctly applied, but several adjusted odds ratios show very wide confidence intervals (e.g., AOR 3.23; 95% CI 2.39–15.60), suggesting sparse data in subgroups or possible overfitting. Please provide information on the number of events per variable included in the models. Categorization of continuous variables such as sleep duration, dietary practices, BMI, and stress requires further justification. The rationale for cutoffs (e.g., ≤5 hours for sleep) should be explained and ideally supported by prior literature. The interpretation of education level as a risk factor is inconsistent: results tables indicate higher odds among nurses with degree-level education, while the discussion suggests the opposite. This discrepancy should be resolved.

Data Availability: The statement that “all data are contained within the manuscript” does not meet PLOS data policy. The tables provide only summary statistics, which do not allow replication of analyses. An anonymized dataset containing individual-level data (demographics, blood pressure, behavioral and occupational variables) should be deposited in a public repository such as Dryad, OSF, or Figshare, or provided as supplementary files. This step is necessary for reproducibility.

The limitations section should be expanded. Restriction of the study to only two tertiary hospitals in an urban area, limiting external validity and generalizability. Heavy reliance on self-reported measures (diet, stress, alcohol consumption, sleep), which introduces recall and social desirability bias. Potential residual confounding from unmeasured variables (e.g., shift type, socioeconomic status, other comorbidities). These points should be explicitly acknowledged.

Writing and Presentation: The manuscript is overall intelligible and written in standard English, but there are minor grammatical errors, occasional awkward phrasing, and some unnecessarily long sentences. A careful language revision would improve clarity. Tables are informative but could be streamlined to enhance readability. For example, grouping categories or collapsing less relevant variables could make the results more concise and accessible.

While the findings demonstrate an important burden of hypertension in this occupational group, they cannot be generalized to all Tanzanian nurses or to the national population. Please reframe conclusions to remain within the scope of the study sample and avoid causal implications. The recommendations for workplace interventions are reasonable but should be presented more cautiously, recognizing the cross-sectional design and context-specific limitations.

Reviewer #4: The manuscript is generally very insightful and addresses a very pertinent and interesting topic. While the overall contribution is noteworthy, there are minor comments I recommend the authors to address:

1. The authors described a physically active nurse as one who have had moderate to vigorous activity for at least 10 minutes, 3-5 days a week. The authors should kindly explain what they meant by moderate activity or vigorous activity, and whether it was self-reported by the study participants or observed the researchers. This will provide comprehensive understanding, interpretability, and reliability of the findings.

2. The authors should report distinctively the number and proportion of HPT that were determine through 'self reporting of HPT history', the current usage of HPT medication, and those from the use of the sphygmomanometer.

3. Kindly state the sample size used for this study under the 'methods and material' section, and how the sample size was determined or calculated.

4. Kindly specify the type of multivariable regression model used in this study rather the generalised statement ("Multivariable regression models adjusted for age, sex and high cholesterol, were then used to assess these associations".)

5. The authors should consider reclassifying the BMI from behavioural factors as BMI is an indictor or measure which could be influenced by not only behaviour but genetic, physiology or metabolism, and even environment. Your research did not specifically determine the predictors of the various BMI values among the nurses.

**Do you want your identity to be public for this peer review?** For information about this choice, including consent withdrawal, please see our Privacy Policy

Reviewer #1: No

Reviewer #2: No

Reviewer #3: **Yes: ** Karla Aguirre Ordóñez

Reviewer #4: No

---

## [Editor Report · Decision Letter 1]

26 Dec 2025

Prevalence of Hypertension and Associated Factors among Nurses Working in Private and Public Tertiary Hospitals in Dar Es Salaam, Tanzania

PGPH-D-25-02168R1

Dear Miss Rutagengwa,

We are pleased to inform you that your manuscript 'Prevalence of Hypertension and Associated Factors among Nurses Working in Private and Public Tertiary Hospitals in Dar Es Salaam, Tanzania' has been provisionally accepted for publication in PLOS Global Public Health.

Best regards,

Suma Krishnasastry, MBBS, MD,DNB, FRCP (Edin)

Academic Editor